## THE NATURAL HISTORY OF MODEL ORGANISMS

# The untapped potential of medaka and its wild relatives

**Abstract** The medaka is a fish that has served as a model organism for over a century, yet there is still much to learn about its life in the wild. Here we summarize the current knowledge, highlight recent progress and outline remaining gaps in our understanding of the natural history of medaka. It has also become clear over time that rather than being a single species, medaka comprises an entire species complex, so disentangling the species boundaries is an important goal for future research. Moreover, medaka and other ricefishes exhibit striking functional diversity, little of which has been investigated to date. As such, there are opportunities to use the resources developed for medaka to study other ricefishes, and to learn more about medaka itself in an evolutionary context.

**LEON HILGERS\* AND JULIA SCHWARZER\***

**Competing interests:** The authors declare that no competing interests exist.

## Introduction

Medaka (*Oryzias latipes* species complex) is a small, egg-laying freshwater fish from East Asia (*Figure 1*). It is often found in rice fields, which earned it its common English name of ricefish and the genus name of *Oryzias* (based on the genus name of rice, *Oryza*). The medaka has been kept as an ornamental fish in Japan from as early as the 17th century, and its central role in Japanese culture is exemplified by its appearance in pieces ranging from Edo era paintings (*Kinoshita et al., 2009*; *Naruse et al., 2011*; *Takeda and Shimada, 2010*) to modern-day children's songs (*Mamun et al., 2016*).

The first medaka species was described as *Poecilia latipes* by *Temminck and Schlegel (1846)* in the *Fauna Japonica* based on collections by the German physician and naturalist Philipp Franz von Siebhold. It was reclassified in the genus *Oryzias* by *Jordan and Snyder (1906)* to become *Oryzias latipes*, and, more recently, it has become clear that the medaka comprises a species complex rather that a single species (*Table 1*).

The traditional breeding of naturally occurring color mutants led to experiments in the early 20th century that sparked the medaka's role as a model organism. It later proved to be an ideal laboratory organism that was easy to maintain due to its small body size, and its simple dietary and habitat requirements. The combination of a sexual dimorphism, short generation times, large and transparent eggs with easily observable development, high tolerance to inbreeding and a comparably small genome size (~700 Mb) also made medaka suitable for a wide range of studies (*Braasch et al., 2015*; *Kasahara et al., 2007*; *Kirchmaier et al., 2015*; *Parenti, 2008*; *Wittbrodt et al., 2002*). Today, the medaka is a fully fledged model vertebrate that is deeply rooted within the life sciences (*Hori, 2011*; *Kirchmaier et al., 2015*; *Shima and Mitani, 2004*; *Takeda and Shimada, 2010*; *Wittbrodt et al., 2002*).

Early systematic crossing of color mutants of medaka provided evidence for Mendelian inheritance in vertebrates, sex limited inheritance and crossing over between sex chromosomes (*Aida, 1921*; *Ishikawa, 1913*; *Toyama, 1916*). Subsequently, the first hormone induced sex-reversal (*Yamamoto, 1958*) and the discovery of the first sex determining gene in a non-mammalian vertebrate (*Matsuda et al., 2002*) paved the way for the medaka as a model for sex determination and sex differentiation (*Kondo et al., 2009*). Scientific milestones achieved with the medaka further include the detection of the first active DNA-based transposable element (*Tol1*) in a vertebrate (*Koga et al., 1995*), which laid

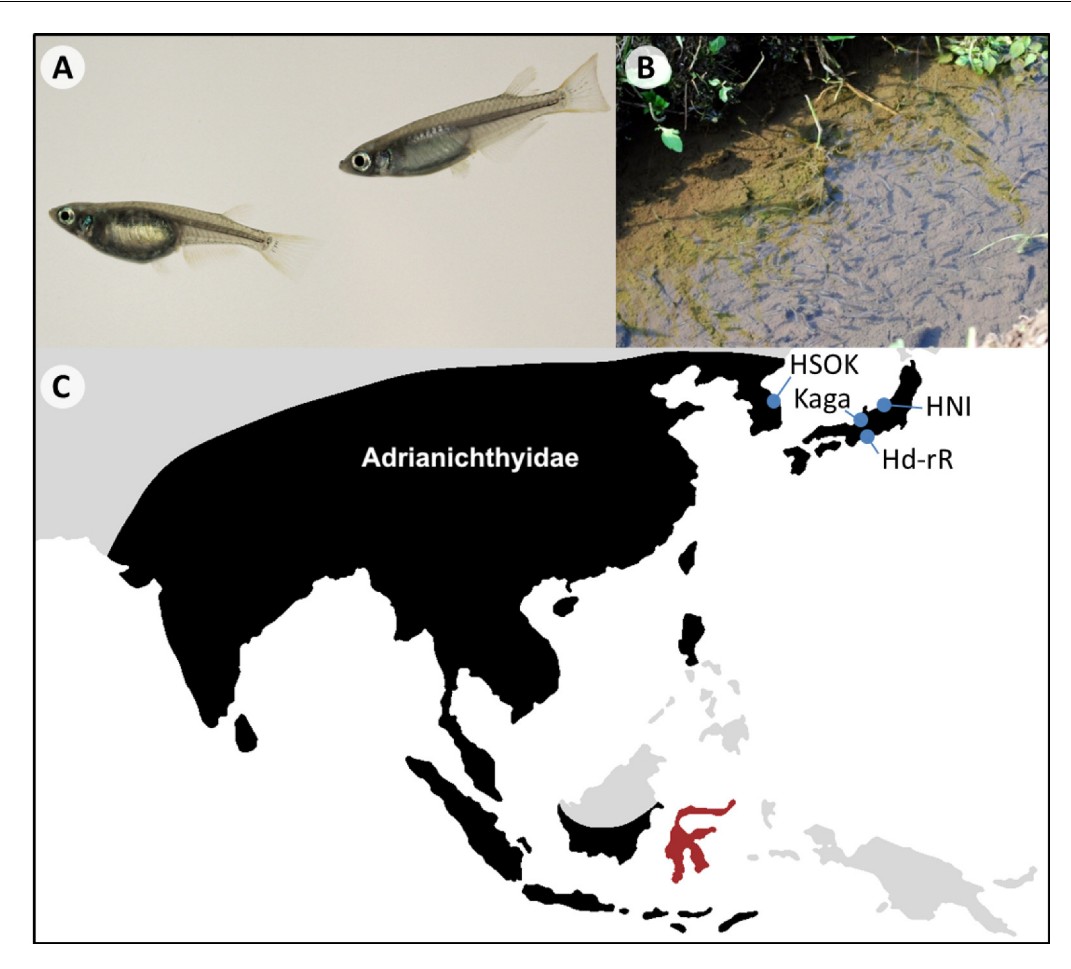

**Figure 1.** *Oryzias latipes.* (**A**) Male (right) and female (left) *Oryzias latipes* from Kiyosu (Photo by Felix Loosli). Males can be easily distinguished from females by their elongated anal and dorsal fins. (**B**) A school of medaka in their natural habitat. (**C**) Approximate distributional limits of the ricefishes family, the Adrianichthyidae (black), following *Parenti (2008)* with the locations from which some of the most famous medaka strains are derived (blue). The Indonesian island Sulawesi represents a ricefish biodiversity hotspot (red).
DOI: https://doi.org/10.7554/eLife.46994.002

the foundation for the development of several tools for genetic modification (*Koga, 2011*).

Over time, medaka developed into a model for development, toxicology, carcinogenesis and behavior and a growing number of inbred strains in combination with an expanding genetic tool-box provided ever growing resources for genetic studies (reviewed in *Kirchmaier et al., 2015*). Medaka's role as genomic model organism was consolidated when its genome became available as one of the first for any vertebrate (*Kasahara et al., 2007*). More recently, the gene-editing tool CRISPR/Cas9 was also established in medaka and employed in multiple studies (e.g. *Gutierrez-Triana et al., 2018*; *Letelier et al., 2018*; *Murakami et al., 2017*; *Watakabe et al., 2018*). In many fields,

including cancer research, medaka serves as complementary model to the well-established zebrafish (*Schartl and Walter, 2016*; *Takeda and Shimada, 2010*). In particular, the natural variation captured in several inbred strains and an isogenic panel offers a unique resource to study allelic variation including disease modifier genes (*Schartl and Walter, 2016*; *Spivakov et al., 2014*).

Due to its long and successful history as a model organism, several comprehensive reviews focus on the medaka and its role in different fields of research (*Kinoshita et al., 2009*; *Kirchmaier et al., 2015*; *Naruse et al., 2011*; *Shima and Mitani, 2004*; *Takeda and Shimada, 2010*; *Wittbrodt et al., 2002*; *Yamamoto, 1975*). In contrast, the natural history of

**Table 1.** Medaka nomenclature.

| Common names | Scientific name |
| --- | --- |
| Japanese medaka, Southern (Japanese) population | O. latipes |
| Northern medaka, Northern (Japanese) population | O. sakaizumii |
| East Korean population | N/A |
| China–West Korean population | O. sinensis |
| Chinese medaka | O. sinensis |
| Taiwanese population, Taiwanese medaka | O. cf. sinensis |

The model organism medaka comprises several species and deeply divergent lineages from the *Oryzias latipes* species complex. Hence, in this article we only use the term "medaka" when we collectively refer to lineages within the *Oryzias latipes* species complex in the context of the model system. Common or scientific species names are used to specifically refer to individual lineages as shown in this table.

DOI: https://doi.org/10.7554/eLife.46994.003

the medaka and its relatives has so far gained comparatively little attention. This is despite the fact that particularly Japanese biologists have been working on ricefishes for decades, which generated a more complete knowledge of *O. latipes* and its relatives than is available for many other fish species (*Iwamatsu, 2006*). However, some of this work is not available in English and thus is less accessible to the international research community. Furthermore, studies have highlighted a so far underexplored ricefish diversity in the wild (*Katsumura et al., 2019*; *Parenti, 2008*; *Takehana et al., 2005*; *Uwa et al., 1988*), which has already inspired new research agendas (e.g., *Matsuda and Sakaizumi, 2016*; *Mokodongan et al., 2018*; *Spivakov et al., 2014*). The combination of available medaka resources and the diversity of its wild relatives represent a treasure trove to target various aspects of biology, which will equally expand our insight into the medaka as a model system. Here, we focus on the natural history of ricefishes and point out recent progress as well as remaining potential for studies focusing on their natural diversity.

## Medaka's life in the wild

Medaka species are distributed in Japan, China, Taiwan and Korea (*Shima and Mitani, 2004*; *Yamamoto, 1975*), where they are mainly found in standing and slow-flowing water bodies such as rice paddies, ponds and agricultural channels (*Fukuda et al., 2005*, *Figure 1B*). The preferred habitats of medaka are shallow zones close to the shoreline with abundant plant cover where it feeds on microscopic organisms, including algae and zooplankton (*Edeline et al., 2016*; *Fukuda et al., 2005*). Although detailed studies on its position in the food web are lacking,

medaka is likely mainly preyed upon by dragonfly larvae, birds and predatory fishes. Throughout its life, medaka is parasitized by both ectoparasites and endoparasites from across the animal kingdom (catalogued in: *Nagasawa, 2017*), and females appear to be particularly susceptible to ectoparasites during the breeding season (*Edeline et al., 2016*). In the wild, most medaka have a lifespan of roughly one year during which males and females can reach a length between three and four centimeters (*Leaf et al., 2011*; *Parenti, 2008*; *Shima and Mitani, 2004*; *Yamamoto, 1975*). In captivity, medaka may live for more than four years (*Egami and Etoh, 1969*). As an inhabitant of the temperate zone, medaka is confronted with and tolerates a wide range of temperatures, from 4°C to 40°C (*Kinoshita et al., 2009*; *Sampetrean et al., 2009*). When the water cools down, fish enter a state of hibernation during which they barely move and stop feeding (*Edeline et al., 2016*; *Kinoshita et al., 2009*). Cooling has been used in the lab to slow down the life cycle of medaka and store it for later use (*Wittbrodt et al., 2002*).

During the breeding season, medaka spawns every morning around sunrise (*Leaf et al., 2011*; *Ueda and Oishi, 1982*). Mating involves an intricate courtship dance during which males repeatedly approach females and exhibit diverse displays (*Ono and Uematsu, 1957*). If accepted, males hold on to the females with their elongated anal and dorsal fins, while females spawn up to 48 eggs (*Leaf et al., 2011*). Bundles of fertilized eggs stay attached to the mother's ovarian cavity via attaching filaments for up to a few hours, before they are deposited (*Parenti, 2005*). Despite its practical use in the laboratory to easily harvest eggs from unambiguously identifiable

## Box 1. Open questions concerning the natural history of ricefishes:

- What is the general ecology of medaka species in the wild: What abiotic factors restrict their distribution; which species do they interact with and in what way?

- How much gene flow exists between *O. latipes* and *O. sakaizumii*?

- Which genes underlie local adaptation in the *O. latipes* species complex?

- How connected are populations of medaka species outside of Japan?

- How much ricefish diversity remains undiscovered?

- What are the major threats to ricefish populations?

- What mechanisms promote evolutionary divergence and functional diversity within medaka and across ricefishes in general?

- What are the molecular mechanisms underlying rapid sex chromosome turnover in ricefishes and what is its role for ricefish diversity?

- What are the processes underlying the remarkable ricefish diversity on Sulawesi, Indonesia?

- Which evolutionary scenarios gave rise to sympatric ricefish species flocks?

- How did pelvic brooding evolve: What is its physiological and genetic basis and what are the adaptive benefits?

DOI: https://doi.org/10.7554/eLife.46994.004

mothers, the reasons for the evolution of this unusual reproductive strategy remain unexplored. Juveniles finally hatch after 7–9 days and grow to maturity within about two and a half months (*Iwamatsu, 2004*; *Takeda and Shimada, 2010*).

Overall, relatively little of our current knowledge on the biology of medaka has been gathered from studies in the wild. Hence, similar to other model organisms, much remains to be discovered about the ecology and life history of wild medaka populations (*Box 1*), and the driving evolutionary forces (*Alfred and Baldwin, 2015*; *Leaf et al., 2011*; *Parichy, 2015*).

## Natural diversity of medaka

### The Oryzias latipes complex

Differences between Japanese medaka populations were recognized early on (*Sakaizumi, 1984*), but it is only now that the scientific community has begun to fully appreciate the diversity summarized within the "medaka". For a long time, the so-called northern and southern Japanese populations, from which the two most famous medaka inbred lines (Hd-rR and HNI) are derived, were regarded as a single species: *Oryzias latipes* (*Figure 1C*). Yet accumulating evidence suggested that these lines represent different species. Hence, the northern Japanese population was described as a new species: *Oryzias sakaizumii* by *Asai et al., 2011*, and medaka is now frequently referred to as "*Oryzias latipes* species complex" (*Iguchi et al., 2018*; *Kawajiri et al., 2015*; *Takehana et al., 2016*). Depending on the method, *Oryzias latipes* and *Oryzias sakaizumii* were estimated to have diverged between 3 and 8 million years ago (*Figure 2A*, *Katsumura et al., 2019*; *Takehana et al., 2003*; but see *Setiamarga et al., 2009*). They differ in a number of traits including craniofacial anatomy (*Kimura et al., 2007*), body coloration (*Asai et al., 2011*), aggressiveness (*Kagawa, 2014*) and the degree of sexual dimorphism of the dorsal fin (*Asai et al., 2011*), the last of which is likely mediated by different sex steroid levels during development (*Katsumura et al., 2014*; *Kawajiri et al., 2015*). Nonetheless, both species produce viable hybrid offspring in the laboratory (e.g. *Murata et al., 2012*) and current evidence also suggests limited gene flow in the wild (*Iguchi et al., 2018*; *Katsumura et al., 2019*). *Oryzias latipes* and *O. sakaizumii* thus offer the opportunity to gain insight into the evolution of physiological and behavioral trait disparity and their genetic bases. Recent insight into the biogeography of the medaka also suggests that *O. latipes* colonized the main island of Japan from the south (i.e., from Kyushu) and subsequently dispersed along the Pacific coast (*Katsumura et al., 2019*). In contrast, *Oryzias sakaizumii* likely originated on

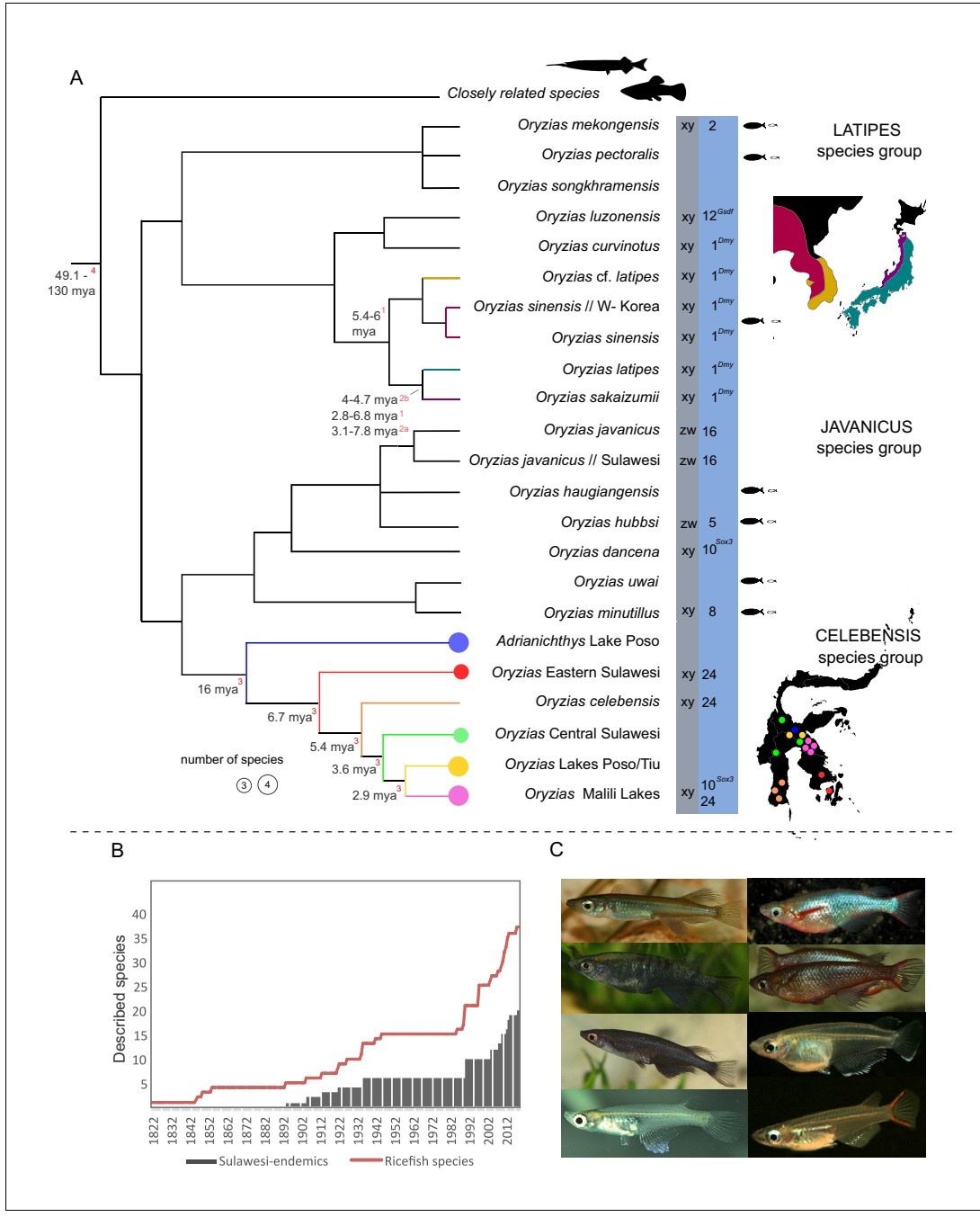

**Figure 2.** Ricefish diversity. (**A**) Combined phylogeny based on published studies from: *Katsumura et al., 2019*, *Parenti, 2008*; *Magtoon, 2010*; *Mokodongan and Yamahira, 2015*; *Takehana et al., 2005*. Sex determination systems and linkage groups with sex determination (SD) genes are given in the gray and blue columns, respectively (based on *Matsuda and Sakaizumi, 2016* and *Myosho et al., 2015*). Species defined as miniaturized (following *Parenti, 2008*) are marked with black/white fish. Detailed distribution maps are provided for the *O. latipes* species complex and for the celebensis species group. Divergence times in the tree are based on: [1]*Takehana et al., 2003*; [2a]*Katsumura et al., 2019*: scenario III; [2b]*Katsumura et al., 2019*: scenario IV; [3]*Mokodongan and Yamahira, 2015*; [4]*Hughes et al., 2018*. (**B**) Total number of described species from the first description of a ricefish in 1822 (*Hamilton, 1822*) until today (red line) in relation to described Sulawesi endemics (gray bars). The high number of newly described species in recent years especially on Sulawesi points towards a largely underexplored ricefish diversity. (**C**) Diversity in coloration and shapes of male *Oryzias*. From left to right and top to bottom: *O. sarasinorum*[*], *O. woworae*[§], *O. eversi*[*], *O. wolasi*[*], *O. nigrimas*[*], *O. dancena*[§], *O. minutillus*[§], *O. mekongensis*[§] (Photos were taken by A Wagnitz[*] and J Geck[§]).

DOI: https://doi.org/10.7554/eLife.46994.007

## Box 2. Pelvic brooding – a complex innovation.

Some ricefishes from Sulawesi exhibit a unique reproductive strategy referred to as "pelvic brooding" (***Kottelat, 1990***). While most ricefishes deposit fertilized eggs shortly after spawning, pelvic brooders instead carry an egg-clutch in a ventral concavity until the fry hatches (***Box 2—figure 1***). Pelvic brooding likely relies on a complex set of adaptations (***Iwamatsu et al., 2007***; ***Iwamatsu et al., 2008***; ***Kottelat, 1990***; ***Parenti, 2008***). Elongated pelvic fins cover the eggs, while a plug structure that is connected to the eggs via attaching filaments anchors them inside the female's ovarian cavity. Ovulation is delayed in pelvic brooders for as long as eggs are attached, and the plug was even assumed to allow the transfer of nutrients, akin to a placenta (***Iwamatsu et al., 2008***). Despite this complexity, pelvic brooding occurs in at least three species from two lineages, namely *Adrianichthys oophorus*, *Oryzias eversi* and *Oryzias sarasinorum*. This means it was either lost multiple times or evolved twice independently (***Mandagi et al., 2018***; ***Mokodongan and Yamahira, 2015***). Pelvic brooding was suggested to have evolved in adaptation to the absence of suitable spawning substrates in pelagic habitats (***Herder et al., 2012a***). The recent discovery of the oviparous *O. dopingdopingensis* by ***Mandagi et al. (2018)***, a sister species of the pelvic brooding *O. sarasinorum* and *O. eversi,* in a river in central Sulawesi might be considered as support for this hypothesis. However, the pelvic brooding *O. eversi* was described from a spring where potential spawning substrates are abundant (***Herder et al., 2012a***). Consequently, the eco-evolutionary dynamics promoting pelvic brooding as well as the underlying molecular mechanisms remain virtually unknown.

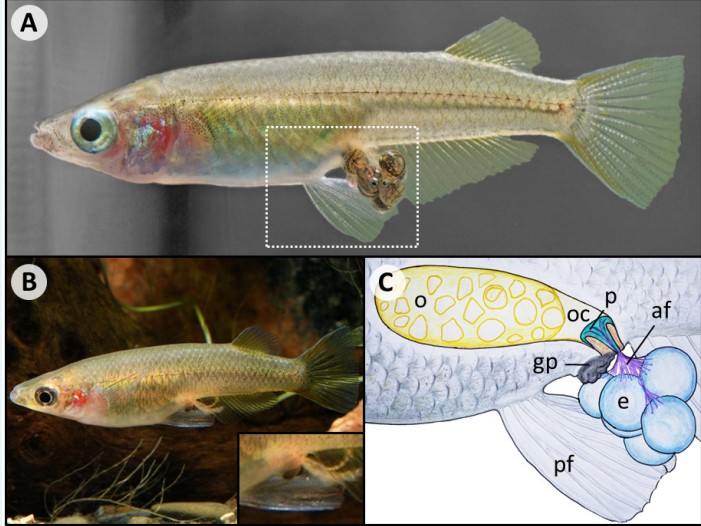

**Box 2—figure 1.** Overview of pelvic brooding.(**A**) The pelvic brooding ricefish *O. eversi* carries its eggs until the fry hatches (dashed box indicates section depicted in (**C**)). (**B**) Once the fry has hatched, only the filaments remain attached to the female (Photo by Hans-Georg Evers). (**C**) A schematic overview of anatomic structures that likely play a role in pelvic brooding. The eggs (e) stay attached to the female via attaching filaments (af). A plug (p) anchors the attaching filaments (af) inside the female's ovarian cavity (oc) next to the ovary (o). Compared to their non-pelvic brooding relatives, pelvic brooding species further appear to have elongated pelvic fins (pf) that cover the eggs and a heavily pigmented, enlarged genital papilla (gp), whose role in pelvic brooding is unclear.

DOI: https://doi.org/10.7554/eLife.46994.006

DOI: https://doi.org/10.7554/eLife.46994.005

the western side of the Japanese Alps, which still largely separate both species in their current distribution (*Figure 2A*, *Katsumura et al., 2019*).

The Korean medaka lineages are genetically divergent from the Japanese species and also highly structured (*Figures 1C* and *2A*). They comprise at least two subgroups that have been referred to as the East Korean (HSOK strain) and China–West Korean populations (*Ds et al., 2013*; *Kang et al., 2005*; *Katsumura et al., 2019*; *Parenti, 2008*). Although still frequently referred to as *O. latipes* (see *Katsumura et al., 2019*; *Spivakov et al., 2014*; *Takeda and Shimada, 2010*), the China–West Korean

population together with populations from Taiwan, likely represents a different species: *Oryzias sinensis* (see *Ds et al., 2013*; *Parenti, 2008*; *Tzeng et al., 2006*). While further data are required, mitochondrial phylogenies indicate the existence of a locally restricted ancient Taiwanese medaka lineage and two lineages derived from more recent invasion events (*Tzeng et al., 2006*). Continental *O. sinensis* are also genetically diverse and geographically distant populations exhibit signatures of divergence, for example, between Shanghai and Korea (*Katsumura et al., 2019*). However, comprehensive studies covering the morphological and genetic diversity in continental Asia are still lacking. Such studies will be quintessential to recognize species limits and distribution patterns in the future. Generally, whether or not divergent lineages within the *O. latipes* species complex represent different species is still treated controversially and inconsistently in the current literature (see *Katsumura et al., 2019*; *Parenti, 2008*; *Sasaki and Yamahira, 2016*; *Spivakov et al., 2014*; *Tzeng et al., 2006*). Aside from the underlying taxonomic debate, information about the degree of divergence and abundance of gene flow between lineages is fundamental for the design of experiments, as well as the interpretation and integration of their results.

### Latitudinal adaptation in Japanese medaka species

The wide range of Japanese medaka species has inspired research on latitudinal adaptation. For example, a recent study identified differences in courtship between populations of *O. sakaizumii*, which were inferred to result from stronger sexual selection at lower latitudes (*Sasaki and Yamahira, 2016*). Additionally, juveniles of *O. sakaizumii* from higher latitudes compensate for a shorter growing season further north with faster growth rates (*Yamahira and Takeshi, 2008*). However, faster growth, which is achieved by higher feeding rates, comes with a cost and is accompanied by higher vulnerability for predation by dragonfly larvae (*Suzuki et al., 2010*). Hence, different growth capacities in *O. sakaizumii* likely evolve in response to both predation pressure and length of the growing season (*Suzuki et al., 2010*). One study also found delayed fin development in northern populations of *O. sakaizumii* compared to the southernmost populations of *O. latipes*, which was speculated to have evolved in a trade-off for faster body growth (*Kawajiri et al., 2009*). However, this study illustrates the caveats related to taxonomic uncertainties, because the authors unintentionally compared populations of different species and thus did not take other interspecific differences into consideration. Nonetheless, their interpretation might be accurate and interspecific diversity clearly represents a chance to gain insight into its evolution, if species boundaries are recognized.

## The diversity of medaka's wild relatives

### Ricefish phylogeny

Although mainly recognized for medaka, ricefishes comprise a considerable diversity with 36 described species in two genera (*Oryzias*: 32 species, *Adrianichthys*: four species, *Asai et al., 2011*; *Herder et al., 2012a*; *Herder and Chapuis, 2010*; *Magtoon, 2010*; *Mandagi et al., 2018*; *Mokodongan et al., 2014*; *Parenti, 2008*; *Parenti et al., 2013*; *Parenti and Hadiaty, 2010*). They form three major clades referred to as the latipes-, javanicus- and celebensis species group, the last of which diversified on Sulawesi, Indonesia (*Figure 2A*, *Mokodongan and Yamahira, 2015*; *Takehana et al., 2005*). With a total of 21 species and 19 local endemics, this island represents a hotspot of ricefish diversity (*Figures 1C* and *2A–B*). Additionally, nine newly described species within the last decade, eight of them endemic to Sulawesi, suggest that both the global ricefish biodiversity as well as the ricefish fauna of Sulawesi are still underdescribed (*Figure 2A*). Nonetheless, the ricefishes known to science exhibit striking functional diversity, with relatively little of it understood or even investigated so far. In the following paragraphs we highlight research making use of ricefish diversity together with aspects that appear especially promising for future research endeavors.

### Sex determination

Sex determination mechanisms are highly variable in bony fishes (*Heule et al., 2014*) and genetic sex determination (SD) appears to be particularly diverse in ricefishes (*Kondo et al., 2009*; *Matsuda and Sakaizumi, 2016*; *Myosho et al., 2015*; *Takehana et al., 2007a*; *Takehana et al., 2007b*). So far, seven sex chromosomes and three different master SD genes have been identified in ricefish species (*Figure 2A*). For example, *O. latipes* has a young XY system in which *dmrt1bY* (*dmy*) on linkage

group 1 (LG1) acts as the male SD gene (*Matsuda, 2005*; *Matsuda et al., 2002*). In contrast, *gsdf* on LG12 is the SD gene in *O. luzonensis* (*Myosho et al., 2012*), and *sox3* on LG10 serves as SD gene in *O. dancena*, *O. marmoratus* and *O. profundicola* (*Figure 2A*, *Myosho et al., 2015*; *Takehana et al., 2014*). *Oryzias hubbsi* and *O. javanicus* even exhibit a ZW, i.e. female heterogametic, system, albeit with different sex chromosomes (*Takehana et al., 2007b*). This astonishing diversity of young sex determination systems inspired ongoing research on the molecular mechanisms underlying rapid sex chromosome turnover and the rewiring of gene regulatory networks that is required to establish new master SD genes (*Herpin et al., 2010*; *Matsuda and Sakaizumi, 2016*).

### Salinity tolerance

Ricefishes live in hyper- and hypoosmotic environments and exhibit different levels of salinity tolerance (*Hayakawa et al., 2015*; *Inoue and Takei, 2003*; *Inoue and Takei, 2002*). While most *Oryzias* are primarily found in freshwater, *O. javanicus* and *O. dancena* (also referred to as *O. melastigma*, likely a synonym; *Parenti, 2008*; *Roberts, 1998*) occur in saltwater and brackish water, respectively. These species are being established as models for marine ecotoxicology to complement the medaka's role in freshwater (*Chen et al., 2011*; *Dong et al., 2014*; *Kang et al., 2013*; *Kim et al., 2018*; *Koyama et al., 2008*). Mobility between freshwater and saltwater depends on an organism's ability to adjust to the osmotic differences. Interestingly, *Oryzias latipes* can better adjust to saltwater when exposed to elevated levels of salinity earlier in life (*Miyanishi et al., 2016*). In contrast, species from some remote freshwater systems in Indonesia do not tolerate even small changes in osmolarity (*Inoue and Takei, 2002*; *Myosho et al., 2018*). The ability of *O. latipes* to acclimate to seawater was used to study the physiological and genetic basis of salt-water tolerance. Salinity tolerance in *Oryzias* is based on both genetic (*Myosho et al., 2018*, *Ogoshi et al., 2015*) and epigenetic factors, including an increase in the density of osmoregulating cells called ionocytes (*Liu et al., 2016*; *Miyanishi et al., 2016*) and an increase in activity of the same cells (*Kang et al., 2008*).

### Sulawesi: A natural laboratory of ricefish diversity

In addition to their species richness, Sulawesi ricefishes stand out due to their diversity in several traits (*Kottelat, 1990*; *Myosho et al., 2018*; *Myosho et al., 2015*; *Parenti, 2008*). While most ricefishes are largely pale in color, Sulawesi ricefishes exhibit several different fin and nuptial colorations and males in the *O. woworae* complex evolved an outstanding iridescent blue ornamentation (*Figure 2C*, *Parenti et al., 2013*). This diversity allows us to expand on the tremendous knowledge of medaka color mutants and to investigate, for example, its genetic basis and the evolutionary role of coloration disparity in ricefishes. Ricefishes further have a wide range of body shapes and sizes (*Parenti, 2008*). Within the *O. woworae* species complex there is variation in body depth (*Parenti et al., 2013*), which likely evolved in response to different flow regimes (*Mokodongan et al., 2018*) and raises the question of interspecific diversity and plasticity of this trait. The largest ricefish *A. poptae* from the ancient Lake Poso reaches more than ten times the size of miniature species such as *O. mekongensis* (see *Parenti, 2008*). Investigating the evolutionary mechanisms that gave rise to the sympatric ricefish species flocks from Lake Poso (namely *Oryzias orthognathus*, *O. nigrimas*, *O. nebulosus* and *Adrianichthys oophorus*, *A. kruyti*, *A. roseni*, *A. poptae*) appears particularly intriguing (*Mokodongan and Yamahira, 2015*). Virtually nothing is known about the ecology and evolution of these species, but striking differences in their craniofacial anatomy point at specialization in their feeding habits as a potential driver of lineage diversification (*Parenti, 2008*; *Parenti and Soeroto, 2004*). Finally, a new reproductive strategy referred to as "pelvic brooding" has evolved in at least two lineages from Sulawesi (*Herder et al., 2012a*; *Kottelat, 1990*). Pelvic brooders do not deposit fertilized eggs, but instead carry their clutch until the fry hatches (*Box 2*, *Iwamatsu, 2004*; *Iwamatsu et al., 2008*; *Parenti, 2008*). While this innovation likely relies on a complex set of behavioral, physiological and anatomical adaptations, the adaptive benefit of this female investment remains unexplored. The potentially repeated and likely relatively recent evolution of pelvic brooding in combination with the available resources for medaka make it a promising system to gain insight into the molecular basis of evolutionary innovations, which is a central goal

of evolutionary biology (*Erwin, 2015*; *Shubin et al., 2009*; *Wagner and Lynch, 2010*).

## New opportunities from technological advances

Technological progress continues to open up new opportunities to study the molecular basis of the natural diversity of ricefishes. Genome evolution can now be studied using long sequencing reads (*Eid et al., 2009*; *Jain et al., 2018*; *Vaser et al., 2017*) to shed light on the contribution of structural variants (see *Sedlazeck et al., 2018*) and methylation patterns (*Simpson et al., 2017*) to adaptation and speciation. The development of single cell transcriptome sequencing also enables insight into cell-specific gene expression patterns (*Andrews and Hemberg, 2018*; *Fiers et al., 2018*; *Tang et al., 2010*). Single cell transcriptome sequencing can, for example, be used to study the evolution of gene regulatory networks underlying sex determination. All of these technologies can help to identify candidate loci that may contribute to evolutionary processes of interest, while the phenotypic effects of natural genetic variation at these loci can now be assessed using gene-editing tools (*Gutierrez-Triana et al., 2018*; *Letelier et al., 2018*; *Murakami et al., 2017*; *Watakabe et al., 2018*). Together these technological advances have set the stage for scientists to begin converting the described natural diversity of ricefishes into a better understanding of the molecular basis of evolutionary processes.

## Threatened diversity

The world's freshwater biota is facing a biodiversity crisis and ricefishes are no exception. Only seven out of 21 assessed ricefish species are classified as least concerned according to the IUCN Red List. However, much of the currently available data needs to be treated with caution because the majority of species assessments based on sufficient data were carried out more than two decades ago (*IUCN, 2019*). What is known is that the high endemism among Sulawesi ricefishes, which are threatened by invasive species, habitat destruction, intensive fishing and pollution, makes them particularly vulnerable for extinction (*Herder et al., 2012b*; *Hilgers et al., 2018*; *Mokodongan et al., 2014*; *Parenti, 2008*). Accordingly, the Lake Poso endemics *A. roseni* and *A. kuyti*, which have not been caught since 1978 and 1983, respectively, are either on the brink of extinction or have already gone extinct (*Kottelat, 1990*). However, severe threats are not restricted to medaka's relatives from Sulawesi. Populations of both Japanese ricefish species have reportedly been declining, particularly in densely populated regions and where exotic species have been introduced (*Asai et al., 2011*; *Mamun et al., 2016*; *Parenti, 2008*; *Parenti and Soeroto, 2004*). Ricefishes were even thought to have gone extinct in Taiwan, before they were rediscovered in 1993 (*Parenti, 2008*; *Tzeng et al., 2006*). In conclusion, current threats and their potential effect often remain speculative. Thus, new species assessments and conservation actions are urgently needed, and ultimately their success will also rely on a better understanding of medaka's life in the wild.

## Conclusions

Despite recent progress, there is much more we can learn about the natural history of medaka. The taxonomic debate around medaka that is fueled by recent insights clearly calls for further investigations, particularly targeting continental medaka species. Combining the tools and resources available for medaka with the diversity of its wild relatives harbors tremendous potential for evolutionary biology. In turn, putting our knowledge on medaka into an evolutionary context may inspire and inform further research in the fields that have traditionally used medaka as a model organism. Sulawesi appears to be an obvious focal point for studying the evolution of ricefish diversity. Conservation efforts targeting some of the ricefish species that are most interesting to science and which appear to be on the brink of extinction are urgently needed. Finally, exciting new research opportunities on medaka arise from the recent establishment of new gene-editing tools and the first isogenic population genetic panel for a vertebrate, which will allow interactions between genotype and environment to be dissected (*Spivakov et al., 2014*).

### Acknowledgements

We thank Fabian Herder, Manfred Schartl, Bernhard Misof, Arne Nolte and the entire AG RICE-FISH for helpful discussions and comments to a previous version of this manuscript. We would further like to express our gratitude to Kiyoshi Naruse, Lynne Parenti and one anonymous reviewer as well as the editors Stuart King and Peter Rodgers, who helped improve this manuscript. Finally, we thank Felix Loosli, Jakob Geck,

Hans-Georg Evers and Andreas Wagnitz for generously providing ricefish photos. This work was supported by the Leibniz association, grant P91/2016 to JS.

**Leon Hilgers** is at the Zoological Research Museum Alexander Koenig, Bonn, Germany

L.Hilgers@leibniz-zfmk.de

 https://orcid.org/0000-0002-3539-2513

**Julia Schwarzer** is at the Zoological Research Museum Alexander Koenig, Bonn, Germany

J.Schwarzer@leibniz-zfmk.de

 https://orcid.org/0000-0001-8242-2845

*Competing interests:* The authors declare that no competing interests exist.

## Funding

| Funder | Grant reference number | Author |
|---|---|---|
| Leibniz Association | P91/2016 | Julia Schwarzer Leon Hilgers |

No external funding was received for this work.

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
