## [Decision Letter]

Thank you for submitting your article "The Natural History of Model Organisms: The untapped potential of medaka's wild relatives" for consideration by *eLife*. Your article has been reviewed by three peer reviewers, and the evaluation has been overseen by two Features Editors at *eLife* (Stuart King and Peter Rodgers). The following individuals involved in review of your submission have also agreed to reveal their identity: Kiyoshi Naruse (Reviewer #2); Lynne Parenti (Reviewer #3).

After a consultation with the reviewers, the editor has drafted this decision to help you prepare a revised submission.

Summary:

This essay is being considered as part of a series of articles on "The Natural History of Model Organisms" (https://elifesciences.org/collections/8de90445/the-natural-history-of-model-organisms). Each article should explain how our knowledge of the natural history of a model organism has informed recent advances in biology, and how understanding its natural history can influence/advance future studies.

This article on the "untapped potential of medaka's wild relatives" sheds interesting light on under-explored resources around the long-established genetic model system: the Japanese ricefish or medaka. The authors swiftly review the history and advantages of medaka as a model system before highlighting the possible application of medaka and its relatives for the context of evolutionary biology. The editors and reviewers share the authors' conviction that this article will interest a wide audience that includes scientists working with medaka in the lab, and evolutionary biologists with a focus on other natural model systems. However, a number of details should be attended to prior to publication.

Essential revisions:

1) The reviewers believe that the revised article could do more to acknowledge prior work involving medaka. In particular, Japanese biologists have been comparing their native ricefish species to those found throughout Asia for decades and, as a result, more is known about Oryzias latipes and its close relatives than about many other species of fishes. For example, Takashi Iwamatsu's multiple editions of The Integrated Book for the Biology of the Medaka (most recent edition 2018) are remarkably detailed and comparative texts available in Japanese. The revision should avoid any phrases that could be misinterpreted as a dismissal of this knowledge (e.g. "surprisingly little is known about the natural history of medaka").

2) To further highlight the potential of this model system, reviewer #1 has suggested adding an additional paragraph that discusses how new technical advances could help to convert the described diversity in the natural resources into a molecular understanding of evolutionary processes, such as adaptation and speciation.

3) The article uses the word "medaka" in several ways: once to mean *Oryzias latipes* (Introduction, second paragraph); a second time to mean *O. latipes* and *O. sakaizumii* (publications on the Japanese Medaka before 2011 referred to general characteristics of both of these species unless the particular population was specified); and a third to mean the *O. latipes* complex (Introduction, first paragraph and subsection “Medaka’s life in the wild”). Further, the article also uses "medaka" as a collective and a singular noun. Though this is a common problem when the meaning of a word is changed, it may confuse readers. All of these species have unique common names, so please revise the article with this in mind. Reviewer #3 suggested adding a table of scientific names along with those common names to help readers sort through this information.

4) Reviewer #2 felt that, while they are described as *Oryzias latipes* from the Aichi Prefecture (Aichi-ken), the fish depicted in Figure 1A-B look more like an orange-red variety rather than the wild type brown. Please consider changing the figure legend to acknowledge this, or finding new photos that more clearly show the brown coloration of wild type medaka.

5) Reviewer #2 also thought that the distribution of the ricefish family (Adrianichthyidae) in Figure 1D was potentially misleading. To our knowledge, there is no information about distribution of these fish inside the Asian continent, especially in the Himalayan area. We are also unaware of any information about the *Oryzias latipes* species complex around Hong Kong (subsection “Medaka’s life in the wild”, first paragraph). The revised article should provide additional references to support these claims, or revise the figures and the text to match the distributions described in previous studies.

6) Box 2's discussion of the pelvic brooding could benefit from the inclusion of the recently described species, *Oryzias dopingdopingensis* [Mandagi et al., 2018]. This species is a sister species of *O. eversi* and *O. sarasinorum*, both of which live in lake habitats and have pelvic fin brooding system. *O. dopingdopingensis*, on the other hand, lives in a river system and deposits its fertilized eggs on substrates around its habitat. The reviewers feel that these details are important in the context of the evolutionary scenario of the pelvic fin brooding system in *Oryzias* in Sulawesi.

---

## [Author Response]

Essential revisions:1) The reviewers believe that the revised article could do more to acknowledge prior work involving medaka. In particular, Japanese biologists have been comparing their native ricefish species to those found throughout Asia for decades and, as a result, more is known about *Oryzias latipes* and its close relatives than about many other species of fishes. For example, Takashi Iwamatsu's multiple editions of The Integrated Book for the Biology of the Medaka (most recent edition 2018) are remarkably detailed and comparative texts available in Japanese. The revision should avoid any phrases that could be misinterpreted as a dismissal of this knowledge (e.g. "surprisingly little is known about the natural history of medaka").

We thank the reviewers for raising our awareness for this issue. To avoid statements that could be misinterpreted as dismissal of knowledge based on prior work, we followed the reviewers’ advice and now reference Iwamatsu’s “The integrated book for the biology of the medaka” and added all further citations mentioned by the reviewers. As suggested by reviewer #3 we further changed: "surprisingly little is known about the natural history of medaka" to “there is much more we can learn about the natural history of medaka”.

Additionally: “Overall, very little of our current knowledge on the biology of medaka has been gathered from studies in the wild.” has been modified to “Overall, relatively little of our current knowledge on the biology of medaka has been gathered from studies in the wild.”

Finally, to further acknowledge previous achievements, yet still underline the contrast between knowledge on the model system and the natural history of ricefishes, we modified the following section, which now reads:

“This is despite the fact that particularly Japanese biologists have been working on ricefishes for decades, which generated a more complete knowledge of *O. latipes* and its relatives than is available for many other fish species (Iwamatsu, 2006). However, some of this work is not available in English and thus remains hardly accessible for the international research community. Furthermore, studies have highlighted a so far underexplored ricefish diversity in the wild (Katsumura et al., 2018; Parenti, 2008; Takehana et al., 2005; Uwa et al., 1988), which has already inspired novel research agendas (e.g., Matsuda and Sakaizumi, 2016; Mokodongan et al., 2018; Spivakov et al., 2014).”

2) To further highlight the potential of this model system, reviewer #1 has suggested adding an additional paragraph that discusses how new technical advances could help to convert the described diversity in the natural resources into a molecular understanding of evolutionary processes, such as adaptation and speciation.

We are grateful for the suggestion of reviewer #1 to add an additional paragraph that discusses how recent technical advances can be used to generate a better understanding of the molecular processes that gave rise to the previously discussed natural diversity of ricefishes. We agree that this may be an interesting addition and thus included a short paragraph that discusses recent advances together with some of the benefits that may be employed to shed light onto the evolution of ricefish diversity.

“New opportunities: Technological advances

Technological progress continues to open up new opportunities to study the molecular basis of the natural ricefish diversity. Genome evolution can now be studied using long sequencing reads (Eid et al., 2009; Jain et al., 2018; Vaser et al., 2017) to shed light on the contribution of structural variants (see Sedlazeck et al., 2018) and methylation patterns (Simpson et al., 2017) to adaptation and speciation in ricefishes. Additionally, the development of single cell transcriptome sequencing enables insight into cell-specific gene expression patterns (Andrews and Hemberg, 2018; Fiers et al., 2018; Tang et al., 2010). Single cell transcriptome sequencing can for example be employed to study the evolution of gene regulatory networks underlying sex determination. All of these technologies can help to identify candidate loci that potentially contribute to evolutionary processes of interest. Intriguingly, phenotypic effects of natural genetic variation at these loci can now be assessed using the gene editing tool CRISPR/Cas9 (Gutierrez-Triana et al., 2018; Letelier et al., 2018; Murakami et al., 2017; Watakabe et al., 2018). Consequently, recent technological advances will facilitate converting the described natural ricefish diversity into a better understanding of the molecular basis of evolutionary processes.

3) The article uses the word "medaka" in several ways: once to mean *Oryzias latipes* (Introduction, second paragraph); a second time to mean *O. latipes* and *O. sakaizumii* (publications on the Japanese Medaka before 2011 referred to general characteristics of both of these species unless the particular population was specified); and a third to mean the *O. latipes* complex (Introduction, first paragraph and subsection “Medaka’s life in the wild”). Further, the article also uses "medaka" as a collective and a singular noun. Though this is a common problem when the meaning of a word is changed, it may confuse readers. All of these species have unique common names, so please revise the article with this in mind. Reviewer #3 suggested adding a table of scientific names along with those common names to help readers sort through this information.

We thank the reviewers for pointing out ambiguity and inconsistency in the nomenclature linked to medaka. We agree that this may cause confusion and have revised the manuscript accordingly. Hence, medaka is only used as singular noun in the revised manuscript. Further, we only use the term “medaka” to collectively refer to lineages within the *Oryzias latipes* species complex in the context of the model system. We stick to common or scientific species names when we specifically refer to individual lineages. Furthermore, to help readers sort through this information, we added Table 1 – Medaka nomenclature, which explains our use of medaka within this manuscript. Additionally, to assist the reader, we follow the recommendation of reviewer #3 and added Table 1 with common and scientific names of species in the *Oryzias latipes* species complex.

The explanatory text in Table 1 now reads:

“The model organism medaka comprises several species and deeply divergent lineages from the *Oryzias latipes* species complex. Hence, in this article we only use the term “medaka” when we collectively refer to lineages within the *Oryzias latipes* species complex in the context of the model system. Common or scientific species names are used to specifically refer to individual lineages as shown in Table 1.”

4) Reviewer #2 felt that, while they are described as *Oryzias latipes* from the Aichi Prefecture (Aichi-ken), the fish depicted in Figure 1A-B look more like an orange-red variety rather than the wild type brown. Please consider changing the figure legend to acknowledge this, or finding new photos that more clearly show the brown coloration of wild type medaka.

We followed reviewer #2’s suggestion and chose a different photograph for which no doubt exists with respect to type or locality. The figure and figure legend have been modified accordingly.

5) Reviewer #2 also thought that the distribution of the ricefish family (Adrianichthyidae) in Figure 1D was potentially misleading. To our knowledge, there is no information about distribution of these fish inside the Asian continent, especially in the Himalayan area. We are also unaware of any information about the *Oryzias latipes* species complex around Hong Kong (subsection “Medaka’s life in the wild”, first paragraph). The revised article should provide additional references to support these claims, or revise the figures and the text to match the distributions described in previous studies.

Although we agree with reviewer #2 in that little is known about the continental distribution of Adrianichthyidae, we prepared the figure in accordance with a figure published in the most comprehensive work, i.e. the revision of Adrianichthyidae by Parenti, 2008. Within this study it is also stated that medaka is “widely distributed throughout eastern China, Hong Kong and Hainan Is.”. However, since we lack definite evidence for the distribution of medaka in Hong Kong and this is not an essential point within this manuscript, we decided to remove “Hong Kong”. To point out the reference we followed in preparing Figure 1C, we now explicitly reference Parenti, 2008, in the figure legend. Additionally, we recognize reviewer #2’s concern and now stress that this is an approximation of distributional limits and not a map of known distribution. This modification was also suggested by reviewer #3. The respective section of the figure legend now reads:

”(C) Approximate distributional limits of Adrianichthyidae (black) following Parenti, 2008, with the locations from which some of the most famous medaka strains are derived (blue).”

6) Box 2's discussion of the pelvic brooding could benefit from the inclusion of the recently described species, *Oryzias dopingdopingensis* [Mandagi et al., 2018]. This species is a sister species of *O. eversi* and *O. sarasinorum*, both of which live in lake habitats and have pelvic fin brooding system. *O. dopingdopingensis*, on the other hand, lives in a river system and deposits its fertilized eggs on substrates around its habitat. The reviewers feel that these details are important in the context of the evolutionary scenario of the pelvic fin brooding system in *Oryzias* in Sulawesi.

We agree with the reviewers that the discovery of *O. dopingdopingensis* is very interesting and may add more detail to the discussion of pelvic brooding. However, *O. eversi* does not occur in a lake habitat (see Herder et al. 2012a). To further discuss the potential implication of the recent description of *O. dopingdopingensis*, but also point out the somewhat contradictory evidence for pelvic brooding as an open water adaptation, we modified the following section. “Pelvic brooding was suggested to have evolved in adaptation to the absence of suitable spawning substrates in pelagic habitats (Herder et al., 2012a). The recent discovery of the oviparous *O. dopingdopingensis* Mandagi et al., (2018), a sister species of the pelvic brooding *O. sarasinorum* and *O. eversi,* in doping-doping river might be considered as support for this hypothesis. However, the pelvic brooding *O. eversi* was also described from a river (Herder et al., 2012a). Consequently, the eco-evolutionary dynamics promoting pelvic brooding as well as the underlying molecular mechanisms remain virtually unknown.”